# HS-SPME-GC–MS Profiling of Volatile Organic Compounds and Polar and Lipid Metabolites of the "Stendesto" Plum–Apricot Kernel with Reference to Its Parents

Dasha Mihaylova [1,*], Aneta Popova [2,*], Ivayla Dincheva [3] and Svetla Pandova [4]

[1] Department of Biotechnology, University of Food Technologies, 4002 Plovdiv, Bulgaria
[2] Department of Catering and Nutrition, University of Food Technologies, 4002 Plovdiv, Bulgaria
[3] Department of Agrobiotechnologies, Agro Bio Institute, Agricultural Academy, 8 Dragan Tsankov Blvd., 1164 Sofia, Bulgaria; ivadincheva@yahoo.com
[4] Department of Breeding and Genetic Resources, Fruit Growing Institute, Agricultural Academy, 4000 Plovdiv, Bulgaria; cler66@abv.bg
* Correspondence: dashamihaylova@yahoo.com (D.M.); popova_aneta@yahoo.com (A.P.)

**Abstract:** Plum–apricot hybrids are the successful backcrosses of plums and apricots. Plums and apricots are well-known and preferred by consumers because of their distinct sensory and beneficial health properties. However, kernel consumption remains limited even though kernels are easily accessible. The "Stendesto" hybrid originates from the "Modesto" apricot and the "Stanley" plum. Kernal metabolites exhibited quantitative differences in terms of metabolites identified by gas chromatography–mass spectrometry (GC–MS) analysis and HS-SPME technique profiling. The results revealed a total of 55 different compounds. Phenolic acids, hydrocarbons, organic acids, fatty acids, sugar acids and alcohols, mono- and disaccharides, as well as amino acids were identified in the studied kernels. The hybrid kernel generally inherited all the metabolites present in the parental kernels. Volatile organic compounds were also investigated. Thirty-five compounds identified as aldehydes, alcohols, ketones, furans, acids, esters, and alkanes were present in the studied samples. Considering volatile organic compounds (VOCs), the hybrid kernel had more resemblance to the plum one, bearing that alkanes were only identified in the apricot kernel. The objective of this study was to investigate the volatile composition and metabolic profile of the first Bulgarian plum–apricot hybrid kernels, and to provide comparable data relevant to both parents. With the aid of principal component analysis (PCA) and hierarchical cluster analysis (HCA), differentiation and clustering of the results occurred in terms of the metabolites present in the plum–apricot hybrid kernels with reference to their parental lines. This study is the first providing information about the metabolic profile of variety-defined kernels. It is also a pioneering study on the comprehensive evaluation of fruit hybrids.

**Keywords:** *Prunus* spp.; fruit; volatile organic compounds (VOCs); principal component analysis; metabolic chemotaxonomy; volatolomics

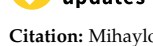



## 1. Introduction

Metabolomic studies are comprehensive tools used to reveal the composition of phytochemicals in various plant tissues and organs [1]. Metabolomics is an omics approach used for more than twenty years in research [2]. It can be divided into targeted and non-targeted approaches [3]. The non-targeted approach detects both known and unknown metabolites, resulting in full profiling [4]. The targeted approach uses a selective known metabolite signal [3]. Metabolomics can use different analytical platforms, including spectroscopy, chromatography, and nuclear magnetic resonance, among others [5]. Metabolomic studies can be used to detect changes during different maturity stages of growing conditions [6]. Gas chromatography–mass spectrometry (GC–MS) is a common technique applied to the

identification and characterization of metabolites' composition. The production of volatile organic compounds (VOCs) is regulated via various metabolomics pathways from their precursors [7]. They are important flavor-contributing agents. Volatile compounds can be divided into primary (synthetized during maturation) and secondary (produced by tissue disruption) ones [8]. Headspace solid-phase microextraction is an effective tool for increased volatile recovery and characterization. It is commonly used with GC–MS. Solid-phase microextraction (SPME)-GC–MS is used in experimental chemistry due to its universalization [9].

The volatile composition of fruits has been widely studied, and different classes of compounds have been documented [10]. However, fruit kernels have not been thoroughly examined in terms of their volatile profile. In fact, the olfactory association of the fruit itself and its kernel widely differs. Kernel oils seldom hold the sensory feeling of the fruit. Consequently, it is of interest to define the major VOCs in kernels as well. The metabolic profile of kernels and the *Prunus* genus in particular is focused mostly on the existence of amygdalin [11]. Its presence is typical for apricot kernels and almonds, and it is known that amygdalin is enzymatically metabolized into cyanide [12]. That is why the quantity of amygdalin is important in terms of preservation of good health. Apricot seeds are also reported to contain several phenolic compounds, i.e., phenolic acids and flavonoids [13]. At present, efforts have been made to achieve sustainable exploitation of resources in every aspect of life. Food provision is viewed as a major societal challenge, and a continuous search for nutritious resources is gaining researchers' interest. The valorization of peels and fruit pomace is not new, and authors have focused on the possibility of incorporating fruit wastes in various industries knowing that they possess many health-enhancing molecules [14,15]. Several papers are now hinting that kernels are rich in metabolites and could be part of the human diet [16]. Some authors suggest that kernels could be valuable supplements in the future due to their beneficial chemical content [17]. Recently, mango seed kernels have been valorized due to their beneficial composition [18].

The plant-based diet has been recognized as having nutritive value. Fruits are generally part of the human diet, and they provide a palette of phytochemicals. Fruits are rich in phytochemicals, i.e., phenolic acids, organic acids, and sugars, among others [19]. It is known that the agrosystem changes gradually due to unfavorable meteorological conditions, the existence of pathogens, or deficiency/toxicity of minerals [20]. The agronomy sector is constantly searching for new sustainable cultivars that have better yield, need less maintenance, and use fewer resources, as well as being able to successfully thrive in the changing climate [21]. The genus *Prunus* has major representatives that are cherished worldwide, like peaches, apricots, plums, and cherries, among others. Fruit hybrids are an interesting approach towards the changing demands of consumers. They combine the most characteristic features of both their parents as well as pose an interesting niche of research with reference to their composition. Plum–apricot fruits are stone fruit like their parents, plums and apricots. Plum–apricot hybrids may result in three main types: plumcots, pluots, and apriums [22]. The "Stendesto" plum–apricot hybrid is the only successful Bulgarian one of the kind, and it is a plumcot. The plumcot is considered 50% plum and 50% apricot. Information about its composition is practically missing; the same applies for its parents, the "Modesto" apricot (father) and the "Stanley" plum (mother). The "Stendesto" plumcot was officially registered in 2013. Not many papers are available on the topic of fruit hybrid composition, which sets new research directions in the identification and application of potential biologically active sources.

A major setback in published papers is the lack of variety/cultivar identification. Not only do the geographical location, soil specificity, and local meteorology factors play an important role in the differences between fruits but also the variety/cultivar itself [23]. This makes it mandatory to pay more attention to the variety differences, especially if they occur in the same species in local and introduced lands. Fruits are providers of vitamins, minerals, and phytochemicals, but they also generate high amounts of by-products

regarding their kernels. Researchers have found ways to incorporate kernels in several industries, i.e., cosmetics, biofuel, detergents, and pharmaceuticals, among others [24]. To date, information about the application of kernels in the food industry is scarce [25]. A thorough holistic approach towards their composition, beneficial compounds, and possible biological activity might set a path for their better understanding and utilization.

The objective of this study was to investigate the volatile composition and metabolic profile of the first Bulgarian plum–apricot hybrid kernels, and to provide comparable data relevant to both parents. With the aid of principal component analysis (PCA) and hierarchical cluster analysis (HCA), differentiation and clustering of the results were provided in terms of the metabolites present in plum–apricot hybrid kernels with reference to their parental lines. This work may not only represent an interesting approach for future studies in line with trending topics like zero-waste management but also be used as core information for further comparison in relative papers. Highlighting kernels as potential nutritional and functional sources will definitely aid in the utilization of this by-product.

## 2. Results and Discussion

The studied kernels were characterized in terms of their VOCs, polar metabolites, and lipids. Information about the compounds found in fruit kernels is scarce or missing. Thus, this is considered a first comprehensive report about the composition of apricot, plum, and plum–apricot kernels. A total of forty-three compounds were identified from the samples (Table 1). Between them, amino acids, organic acids, sugar acids and alcohols, mono- and disaccharides, phenolic acids, and hydrocarbons were identified as existing groups.

**Table 1.** Metabolites (mg/g dry weight) identified in studied kernels analyzed by HS-SPME-GC–MS.

| RI | Class/Name | Modesto | Stanley | Stendesto |
|---|---|---|---|---|
| | **Amino acids** | | | |
| 1105 | Alanine | 0.35 ± 0.11 [a] | 0.22 ± 0.07 [a] | 0.30 ± 0.13 [a] |
| 1232 | Valine | 0.17 ± 0.06 [a] | 0.14 ± 0.05 [a] | 0.08 ± 0.02 [a] |
| 1259 | Leucine | 0.76 ± 0.24 [a] | 0.57 ± 0.18 [ab] | 0.28 ± 0.09 [b] |
| 1296 | Isoleucine | 0.30 ± 0.10 [a] | 0.21 ± 0.07 [a] | 0.12 ± 0.04 [a] |
| 1302 | Proline | 0.93 ± 0.30 [a] | 0.18 ± 0.06 [b] | 0.13 ± 0.04 [b] |
| 1343 | Serine | 1.07 ± 0.34 [a] | 0.73 ± 0.23 [ab] | 0.30 ± 0.10 [b] |
| 1362 | Threonine | 0.25 ± 0.08 [a] | 0.45 ± 0.15 [a] | 0.36 ± 0.12 [a] |
| 1502 | Aspartic acid | 0.78 ± 0.25 [a] | 0.33 ± 0.11 [a] | 0.48 ± 0.15 [a] |
| 1519 | Pyroglutamic acid | 0.64 ± 0.20 [a] | 0.42 ± 0.13 [ab] | 0.11 ± 0.03 [b] |
| 1625 | Phenylalanine | 0.18 ± 0.06 [ab] | 0.37 ± 0.12 [a] | 0.06 ± 0.02 [b] |
| 1656 | Asparagine | 0.68 ± 0.22 [a] | 0.39 ± 0.13 [a] | 0.31 ± 0.10 [a] |
| 1775 | Glutamine | 0.30 ± 0.10 [a] | 0.22 ± 0.07 [a] | 0.16 ± 0.05 [a] |
| 1839 | Arginine | 0.88 ± 0.28 [b] | 5.43 ± 1.74 [a] | 4.61 ± 1.48 [a] |
| | **Organic acids** | | | |
| 1119 | Oxalic acid | 0.73 ± 0.23 [a] | 0.45 ± 0.14 [ab] | 0.11 ± 0.04 [b] |
| 1314 | Succinic acid | 0.19 ± 0.06 [a] | 0.08 ± 0.03 [b] | 0.05 ± 0.02 [b] |
| 1330 | Fumaric acid | 0.11 ± 0.03 [a] | 0.05 ± 0.02 [b] | 0.03 ± 0.01 [b] |
| 1475 | Mallic acid | 1.92 ± 0.62 [a] | 0.05 ± 0.02 [b] | 2.60 ± 0.83 [a] |
| 1530 | γ-Aminobutyric acid | 0.13 ± 0.04 [ab] | 0.16 ± 0.05 [a] | 0.06 ± 0.02 [b] |
| 1727 | 2-Aminoadipic acid | 1.85 ± 0.59 [a] | 1.36 ± 0.44 [a] | 1.07 ± 0.34 [a] |
| 1816 | Isocitric acid | 0.39 ± 0.12 [ab] | 0.24 ± 0.08 [b] | 0.74 ± 0.24 [a] |
| | **Sugar acids and alcohols** | | | |
| 1264 | Glycerol | 1.75 ± 0.56 [a] | 0.36 ± 0.11 [b] | 0.47 ± 0.15 [b] |
| 1541 | Eritrreonic acid | 1.19 ± 0.38 [a] | 0.93 ± 0.30 [a] | 0.52 ± 0.17 [a] |
| 1611 | Glutamic acid | 0.43 ± 0.14 [a] | 0.14 ± 0.04 [b] | 0.15 ± 0.05 [b] |
| 1695 | Xylitol | 2.30 ± 0.74 [a] | 1.65 ± 0.53 [a] | 0.87 ± 0.28 [a] |
| 1718 | Arabitol | 0.68 ± 0.22 [a] | 0.50 ± 0.16 [a] | 0.27 ± 0.09 [a] |
| 1801 | Glyceric acid-3-phosphate | 0.21 ± 0.07 [a] | 0.19 ± 0.06 [a] | 0.43 ± 0.14 [a] |

**Table 1.** *Cont.*

| RI | Class/Name | Modesto | Stanley | Stendesto |
|---|---|---|---|---|
| | | **Sugar acids and alcohols** | | |
| 1920 | Sorbitol | 7.22 ± 2.32 [a] | 12.31 ± 3.95 [a] | 11.53 ± 3.70 [a] |
| 2009 | Gluconic acid | 0.12 ± 0.04 [b] | 0.30 ± 0.10 [a] | 0.11 ± 0.03 [b] |
| 2018 | Glucaric acid | 0.25 ± 0.08 [a] | 0.15 ± 0.05 [a] | 0.12 ± 0.04 [a] |
| 2041 | Myo-Inositol isomer | 0.83 ± 0.27 [a] | 0.53 ± 0.17 [ab] | 0.29 ± 0.09 [b] |
| 2101 | Myo-Inositol isomer | 2.05 ± 0.66 [a] | 0.21 ± 0.07 [b] | 0.74 ± 0.24 [b] |
| | | **Mono- and disaccharides** | | |
| 1855 | Fructose isomer | 2.38 ± 0.76 [b] | 6.48 ± 2.08 [ab] | 7.05 ± 2.11 [a] |
| 1869 | Fructose isomer | 1.60 ± 0.51 [b] | 4.95 ± 1.59 [ab] | 5.51 ± 1.77 [a] |
| 1876 | 1-Methyl-$\alpha$-D-glucopyranoside | 0.10 ± 0.03 [b] | 0.27 ± 0.09 [ab] | 0.36 ± 0.12 [a] |
| 1882 | Glucose isomer | 2.78 ± 0.89 [b] | 9.90 ± 2.18 [a] | 12.00 ± 2.85 [a] |
| 1898 | Glucose isomer | 0.83 ± 0.27 [b] | 2.21 ± 0.71 [ab] | 2.60 ± 0.83 [a] |
| 1937 | Glucose 1-phosphate | 0.16 ± 0.05 [b] | 0.25 ± 0.08 [b] | 4.05 ± 1.30 [a] |
| 2687 | Sucrose | 15.54 ± 2.99 [a] | 7.39 ± 1.37 [b] | 10.25 ± 2.29 [ab] |
| | | **Phenolic acids** | | |
| 1835 | Protocatechuic acid | 0.32 ± 0.10 [ab] | 0.17 ± 0.06 [b] | 0.60 ± 0.19 [a] |
| 1940 | *trans-p*-Coumaric acid | 0.55 ± 0.18 [a] | 0.41 ± 0.13 [a] | 0.24 ± 0.10 [a] |
| 2106 | *trans*-Ferulic acid | 0.19 ± 0.07 [a] | 0.38 ± 0.12 [a] | 0.26 ± 0.08 [a] |
| | | **Others** | | |
| 1400 | Tetradecane | 0.64 ± 0.21 [a] | 0.46 ± 0.15 [ab] | 0.13 ± 0.04 [b] |
| 1600 | Hexadecane | 0.29 ± 0.09 [a] | 0.20 ± 0.06 [a] | 0.12 ± 0.04 [a] |

Amino acids marked in blue color are essential; RI—retention index. Different letters in the same row indicate statistically significant differences ($p < 0.05$) according to ANOVA and the Tukey test.

Amino acids were found in small quantities, yet the hybrid kernel had managed to keep all the amino acids present in both apricot and plum. The same trend applied to the other identified classes, where the hybrid had inherited all the metabolites present in its parents. A study about apricot fruit and seeds showed a resemblance in the metabolites found in the apricot kernel, although only 36 metabolites were identified [26]. Arginine was the most abundant in both hybrid and plum kernels, while serine had the highest value in the apricot kernel. Arginine is important to human nutrition since research has shown that it can increase lipolytic enzymes' activity and decrease insulin resistance [27]. Serine is also reported as exceptionally important, especially being a substrate for glucose and protein synthesis and building of phospholipids [28]. Mallic and 2-aminoadipic acids were the organic acids with the highest values in the studied kernels. The plum–apricot hybrid had the most malic acid, while the apricot had the most 2-aminoadipic acid. Aminoadipic acid is an object of scientific research due to its recent identification as a biomarker of insulin resistance and obesity [29]. Mallic acid is commonly found in fruits, and it is reported to be an antimicrobial agent combined with citric acid [30]. The current results show that organic acids also accumulate in kernels. Sorbitol had the highest values from the group of sugar acids and alcohols. The sorbitol pathway is a two-step one, where in the first step glucose is converted into sorbitol, and then in the second step sorbitol is converted into fructose [31]. The plum–apricot hybrid's kernel had lower quantities of sorbitol compared to its parents but had accumulated more glucose and fructose as isomers. Sorbitol is an alternative sweetener that is widespread in some *Prunus* spp. [32]. Sucrose and glucose and fructose isomers were predominant in the plum–apricot hybrid kernels. The amount of sucrose in the apricot kernel was 50% higher compared to the hybrid. Kernels from Turkish apricots were also high in fructose, glucose, sucrose, and maltose [33]. Protocatechuic acid was the dominant one in the plum–apricot hybrid kernels, while *trans-p*-coumaric acid had its highest values in both its parents. Protocatechuic acid is reported to possess an assortment of biological activities, i.e., antibacterial, antiviral, anticancer, antiosteoporotic, and antioxidant, among others [34].

The overall distribution of the different classes of metabolites is presented in Figure 1.

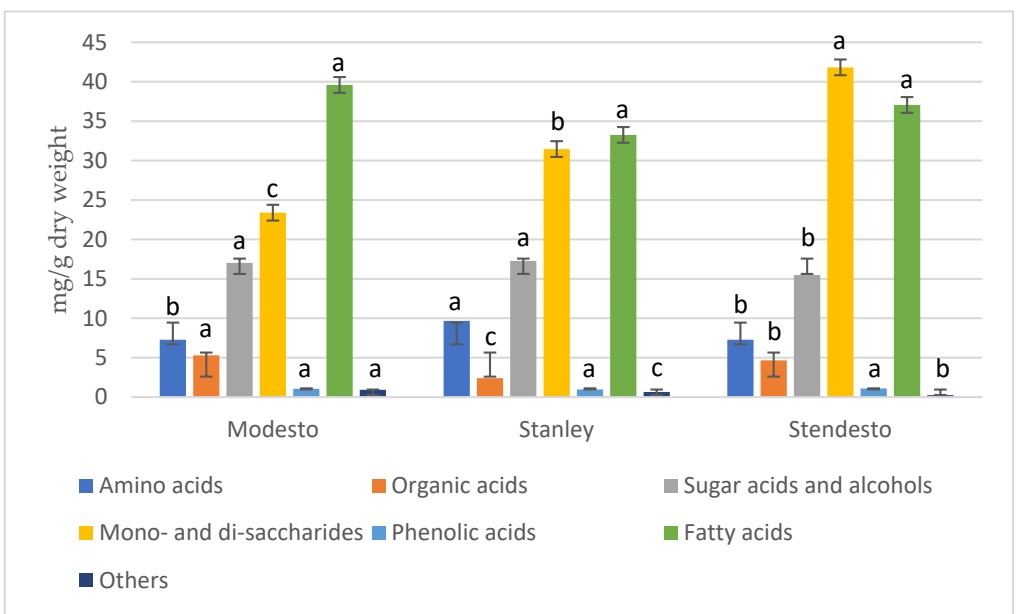

**Figure 1.** Distribution of metabolites in studied fruit kernels according to their chemical families. Different letters in the same chemical family indicate statistically significant differences ($p < 0.05$) according to ANOVA and the Tukey test.

The plum–apricot hybrid kernels had more mono- and disaccharides and phenolic acids compared to the parents, the same amount of amino acids as apricot kernels, and decreased sugar acids and alcohols. Considering the amount of fatty and organic acids, the amount was more similar to the apricot kernels than to the plum kernels. The distribution of the individual compounds was different, and their amounts contributed differently not only to the specific chemical family but also to the kernel variety. Due to the lack of relevant data, comparison with subject reference to other papers cannot be conducted.

The fatty acids content is presented in Table 2, where a total of twelve compounds were identified, including four fatty alcohols. Although more saturated fatty acids were discovered, their amount did not exceed the value of unsaturated fatty acids. The polyunsaturated fatty acids were more frequent, but the amount of monounsaturated fatty acids was greater. The saturated/unsaturated ratios in the studied kernels are as follows: 0.93 ("Modesto"), 1.01 ("Stanley"), and 0.96 ("Stendesto").

**Table 2.** Fatty acids (mg/g dry weight) identified in studied kernels analyzed by HS-SPME-GC–MS.

| RI | Fatty Acids | "Modesto" | "Stanley" | "Stendesto" |
|---|---|---|---|---|
| 1519 | Lauric acid | 3.93 ± 1.12 [a] | 3.16 ± 1.12 [a] | 2.67 ± 1.12 [a] |
| 1572 | Dodecanol | 1.59 ± 0.45 [a] | 1.04 ± 0.45 [a] | 0.97 ± 0.45 [a] |
| 1725 | Mirystic acid | 8.75 ± 2.51 [a] | 8.00 ± 2.51 [a] | 8.66 ± 2.51 [a] |
| 1874 | Tetradecanol | 0.33 ± 0.09 [a] | 0.21 ± 0.09 [a] | 0.24 ± 0.09 [a] |
| 1920 | Palmitic acid | 1.16 ± 0.33 [a] | 0.94 ± 0.33 [a] | 1.07 ± 0.33 [a] |
| 1943 | Hexadecanol | 0.48 ± 0.14 [a] | 0.34 ± 0.14 [a] | 0.64 ± 0.14 [a] |
| 2094 | Linoleic acid | 7.06 ± 2.02 [a] | 4.27 ± 2.02 [a] | 5.08 ± 2.02 [a] |
| 2101 | Oleic acid | 12.68 ± 3.63 [a] | 11.00 ± 3.63 [a] | 13.39 ± 3.63 [a] |
| 2106 | Linolenic acid | 0.78 ± 0.22 [ab] | 1.24 ± 0.22 [a] | 0.38 ± 0.22 [b] |
| 2128 | Stearic acid | 0.94 ± 0.27 [b] | 1.41 ± 0.27 [ab] | 1.71 ± 0.27 [a] |
| 2157 | Octadecanol | 0.89 ± 0.25 [a] | 1.12 ± 0.25 [a] | 1.44 ± 0.25 [a] |
| 2311 | Eicosanoic acid | 1.00 ± 0.29 [a] | 0.52 ± 0.29 [a] | 0.79 ± 0.29 [a] |

RI—retention index. Different letters in the same row indicate statistically significant differences ($p < 0.05$) according to ANOVA and the Tukey test.

Myristic, linoleic, and oleic acids were the predominant ones. Linoleic and oleic acids have been reported as promising anti-mycobacterial agents with high antioxidant

potential [35]. It can be seen that the hybrid inherited the presence of all the identified fatty acids from its parents. Only the amounts of dodecanol and linoleic acids in the plum–apricot hybrid's kernels were less than identified in both plum and apricot. On the other hand, the plum–apricot kernel accumulated more octadecanol, oleic, and stearic acids than both its parents. Other papers investigated the composition of the oil from plum and apricot kernels, where they identified ten fatty acids with the prevalence of oleic and linoleic acids [36], demonstrating consistency with the current results.

The investigated volatile compounds are presented in Table 3. The HS-SPME-GC–MS analysis revealed the existence of thirty-five compounds profiled as aldehydes, alcohols, ketones, furans, acids, esters, and alkanes. While most metabolites were present in all the samples, alkanes were only identified in the apricot kernel. This hinted that the volatile profile of the hybrid kernel was more similar to the plum than to the apricot in terms of class distinction.

**Table 3.** Identified volatile compounds (% of total ion current) in studied kernels analyzed by HS-SPME-GC–MS.

| RI | Name/Class | Modesto | Stanley | Stendesto |
|---|---|---|---|---|
| | **Aldehydes** | | | |
| 566 | 2-Methylpropanal | 5.43 ± 0.91 [a] | 2.50 ± 0.42 [b] | 6.34 ± 1.07 [a] |
| 595 | n-Butanal | 1.52 ± 0.26 [ab] | 1.94 ± 0.33 [a] | 1.22 ± 0.20 [b] |
| 653 | 3-Methylbutanal | 0.61 ± 0.10 [a] | 0.53 ± 0.09 [a] | 0.49 ± 0.08 [a] |
| 667 | 2-Methylbutanal | 2.98 ± 0.50 [a] | 1.33 ± 0.22 [a] | 2.38 ± 0.40 [a] |
| 698 | Pentanal | 4.41 ± 0.74 [b] | 8.12 ± 1.36 [a] | 5.53 ± 0.93 [ab] |
| 752 | (E)-2-Pentenal | 0.88 ± 0.15 [a] | 0.27 ± 0.05 [a] | 0.70 ± 0.12 [a] |
| 792 | n-Hexanal | 9.74 ± 1.64 [b] | 18.61 ± 3.13 [a] | 14.79 ± 2.49 [ab] |
| 830 | 2-Furfural | 2.61 ± 0.44 [a] | 3.40 ± 0.57 [a] | 2.09 ± 0.35 [a] |
| 902 | Heptanal | 1.46 ± 0.25 [b] | 1.67 ± 0.28 [a] | 1.17 ± 0.20 [b] |
| 961 | (E)-2-Heptenal | 6.15 ± 1.03 [a] | 10.19 ± 1.71 [a] | 4.92 ± 0.83 [a] |
| 975 | Benzaldehyde | 26.99 ± 4.53 [b] | 20.82 ± 3.50 [a] | 22.59 ± 3.79 [b] |
| 1011 | n-Octanal | 1.37 ± 0.23 [b] | 2.29 ± 0.39 [a] | 1.10 ± 0.18 [ab] |
| 1073 | (E)-2-Octenal | 3.18 ± 0.53 [a] | 5.34 ± 0.90 [a] | 4.55 ± 0.76 [a] |
| 1106 | *n*-Nonanal | 4.10 ± 0.69 [ab] | 2.84 ± 0.48 [a] | 3.28 ± 0.55 [b] |
| 1146 | (E)-2-Nonenal | 0.71 ± 0.12 [b] | 0.94 ± 0.16 [b] | 0.57 ± 0.10 [a] |
| 1232 | (E)-2-Decenal | 1.08 ± 0.18 [b] | 0.80 ± 0.13 [a] | 1.86 ± 0.31 [a] |
| | **Alcohols** | | | |
| 500 | Ethanol | 0.20 ± 0.03 [b] | 1.00 ± 0.17 [ab] | 1.25 ± 0.21 [a] |
| 680 | 1-Butanol | 0.64 ± 0.11 [b] | 0.95 ± 0.16 [a] | 1.19 ± 0.20 [a] |
| 689 | 1-Penten-3-ol | 0.17 ± 0.03 [ab] | 0.52 ± 0.09 [b] | 0.65 ± 0.11 [a] |
| 770 | 1-Pentanol | 4.25 ± 0.71 [a] | 2.66 ± 0.45 [b] | 5.33 ± 0.90 [b] |
| 1036 | Benzyl alcohol | 10.35 ± 1.74 [a] | 2.30 ± 0.39 [a] | 2.87 ± 0.48 [a] |
| 1173 | 4-Ethylphenol | 0.65 ± 0.11 [a] | 0.48 ± 0.08 [a] | 0.60 ± 0.10 [a] |
| | **Ketones** | | | |
| 515 | 2-Propanone | 0.95 ± 0.16 [a] | 0.44 ± 0.07 [b] | 0.75 ± 0.13 [ab] |
| 691 | 2-Pentanone | 0.75 ± 0.13 [a] | 0.49 ± 0.08 [a] | 0.61 ± 0.10 [a] |
| 892 | 2-Heptanone | 1.58 ± 0.27 [a] | 1.86 ± 0.31 [a] | 2.38 ± 0.40 [a] |
| | **Furans** | | | |
| 995 | 2-Pentylfuran | 1.33 ± 0.22 [b] | 3.89 ± 0.65 [a] | 4.86 ± 0.82 [ab] |
| | **Acids** | | | |
| 741 | Acetic acid | 0.25 ± 0.04 [b] | 2.02 ± 0.34 [a] | 2.74 ± 0.46 [a] |
| | **Esters** | | | |
| 617 | Ethyl acetate | 0.35 ± 0.06 [b] | 0.57 ± 0.10 [a] | 0.12 ± 0.02 [c] |
| 1161 | Benzyl acetate | 0.49 ± 0.08 | ND | 0.69 ± 0.12 |
| 1175 | Ethyl benzoate | 0.63 ± 0.11 | ND | 0.80 ± 0.13 |

**Table 3.** *Cont.*

| RI | Name/Class | Modesto | Stanley | Stendesto |
|---|---|---|---|---|
| | | **Alkanes** | | |
| 900 | Nonane | 0.95 ± 0.16 | ND | ND |
| 1000 | Decane | 0.48 ± 0.08 | ND | ND |
| 1100 | Undecane | 0.67 ± 0.11 | ND | ND |
| 1200 | Dodecane | 0.29 ± 0.05 | ND | ND |
| 1300 | Tridecane | 0.73 ± 0.12 | ND | ND |

RI—retention index; ND—not detected. Different letters in the same row indicate statistically significant differences ($p < 0.05$) according to ANOVA and the Tukey test.

Figure 2 reveals the differences between the hybrid kernels and those of plum and apricot in terms of %TIC (total ion current) predominance and variety dependance.

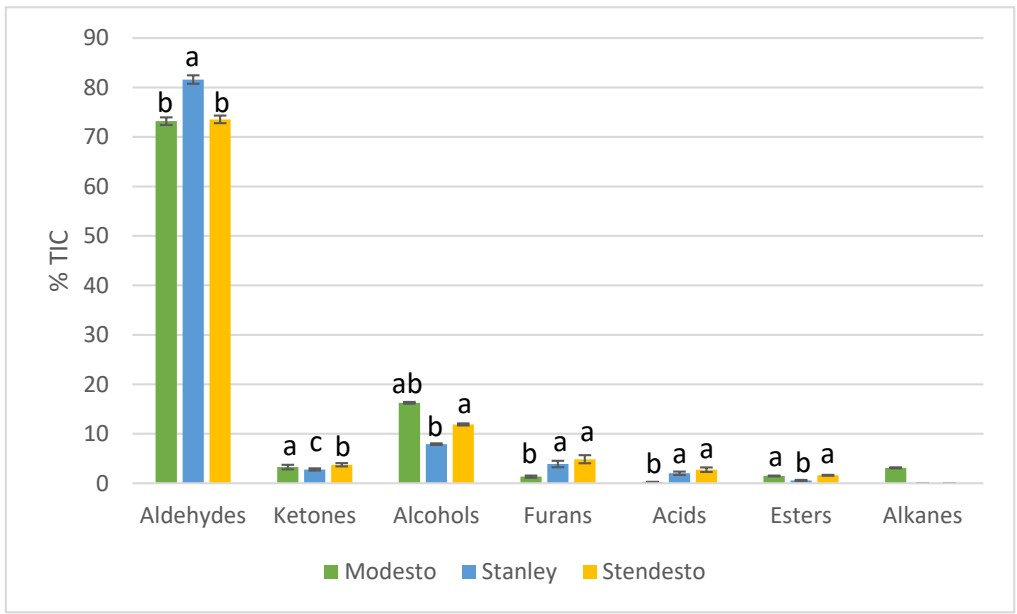

**Figure 2.** Distribution of volatile compounds according to their chemical families in studied fruit kernels. Different letters in the same chemical family indicate statistically significant differences ($p < 0.05$) according to ANOVA and the Tukey test.

Aldehydes contributed the most to the volatile profile of the three kernels. In terms of aldehydes, ketones, and esters, the hybrid showed more similarity to the apricot, while the %TIC of identified furans, alkanes, and acids corresponded more to the plum. The percentage of TIC of the alcohols was less than the ones in the apricot kernel and more than in the plum kernel.

Aroma has always been important not only to the food industry but also to cosmetology, pharmacology, and others. Kernel oils have a distinct smell that is not connected to the olfactory association of the fruit. The most important from the currently identified aldehydes were 2-methylpropanal ("Stendesto"); pentanal, n-hexanal, (E)-2-heptenal ("Stanley"), and benzaldehyde ("Modesto"). The hybrid kernel had the most n-hexanal and benzaldehyde. The least present from the %TIC were 3-methylbutanal and (E)-2-nonenal. Benzaldehyde is connected to the almond aroma [37]. Esters, acids, and alkanes were identified in relatively small %TIC. However, the hybrid kernel held the highest %TIC for acetic acid and ethyl benzoate. Esters contribute to the typical floral and fruity flavor of products [38]; thus, the hybrid kernel should exhibit more floral and fruity volatiles compared to its parents. Acetic acid, on the other hand, is a major odor-active component identified in fruit vinegars [39].

Benzyl alcohol dominated in the apricot kernel, while 1-pentanol dominated in the hybrid one. Benzyl alcohol is associated with a floral odor and a marzipan-like flavor [40] and 1-pentanol with a fruity odor [41]. 2-heptanone was the major ketone in the hybrid kernel. According to research, it contributes to an oxidative odor [42]. The overall volatile assessment can be linked to the threshold levels the different compounds possess. Aldehydes have lower sensory thresholds compared to alcohols [43]; thus, they might be viewed as the largest contributors to the sensory associations of the three studied kernels. Furan and its derivatives usually occur in heat-processed foods and beverages [44]. Moreover, 2-penthylfuran has been identified in a number of foods, i.e., baby food, deep-fried foods, and fruit juice [45], and 2-penthylfuran is the only furan identified in the three kernels, where the hybrid one held the highest %TIC. This compound is known for its distinct fruity flavor and caramel undertones [46].

Figure 3 is a visual presentation of the odor description of each kernel variety based on the VOCs present in them.

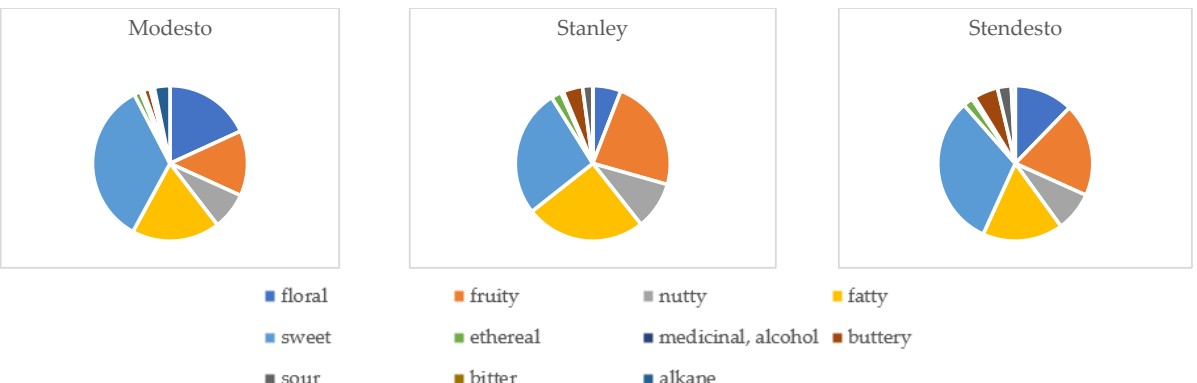

**Figure 3.** Odor component distribution in studied fruit kernels (according to https://foodb.ca/compounds descriptor (accessed 1 December 2023).

It can be seen that the hybrid kernel had distinct sweet (32%) and fatty (17%) sensory properties, while the plum kernel had a more fruity (23%), sweet (27%), and fatty (25%) odor description. The apricot kernel also presented a sweet (34%), fatty (19%), and floral (18%) profile. The least recognized odors in the apricot kernel were the ethereal and bitter ones, while in the plum and plum–apricot kernels those were the ethereal and sour ones. The nutty odor was evenly distributed in the hybrid and plum kernels, and more distinct in the apricot kernel. In terms of sensory perception, the plum–apricot kernel is more similar to the apricot kernel than to the plum kernel.

This research can be viewed as a pioneering study on the topic of metabolite identification of "Modesto" (apricot), "Stanley" (plum), and "Stendesto" (plum–apricot hybrid) kernels and provide a stepping stone for future evaluations and comparisons.

*Principal Component and Hierarchical Cluster Analyses of HS-SPME-GC–MS Data*

The chemical composition and volatile content were analyzed using principal component analysis (PCA) and further explored to distinguished separate groups with hierarchical cluster analysis (HCA). As shown in Figure 4A (metabolites), two principal components were generated in the PCA with an eigenvalue greater than 1, accounting for 65.8% (65% for PC 1 and 35% for PC2) of the total variance, whereas, in Figure 4B (VOCs), 59.7% were distributed for PC1 (52.1%) and PC2 (47.9%). Glucaric acid, glucose-1-phosphate, isocitric acid, 2-heptanone, 1-butanol, and acetic acid are positioned most positively, whereas trans-ferulic acid, threonine, (e)-2-heptenal, and 1-pentanol contributed most negatively.

As illustrated in Figure 5, the samples can be divided into two clusters for both metabolites (Figure 5A) and volatile compounds (Figure 5B).

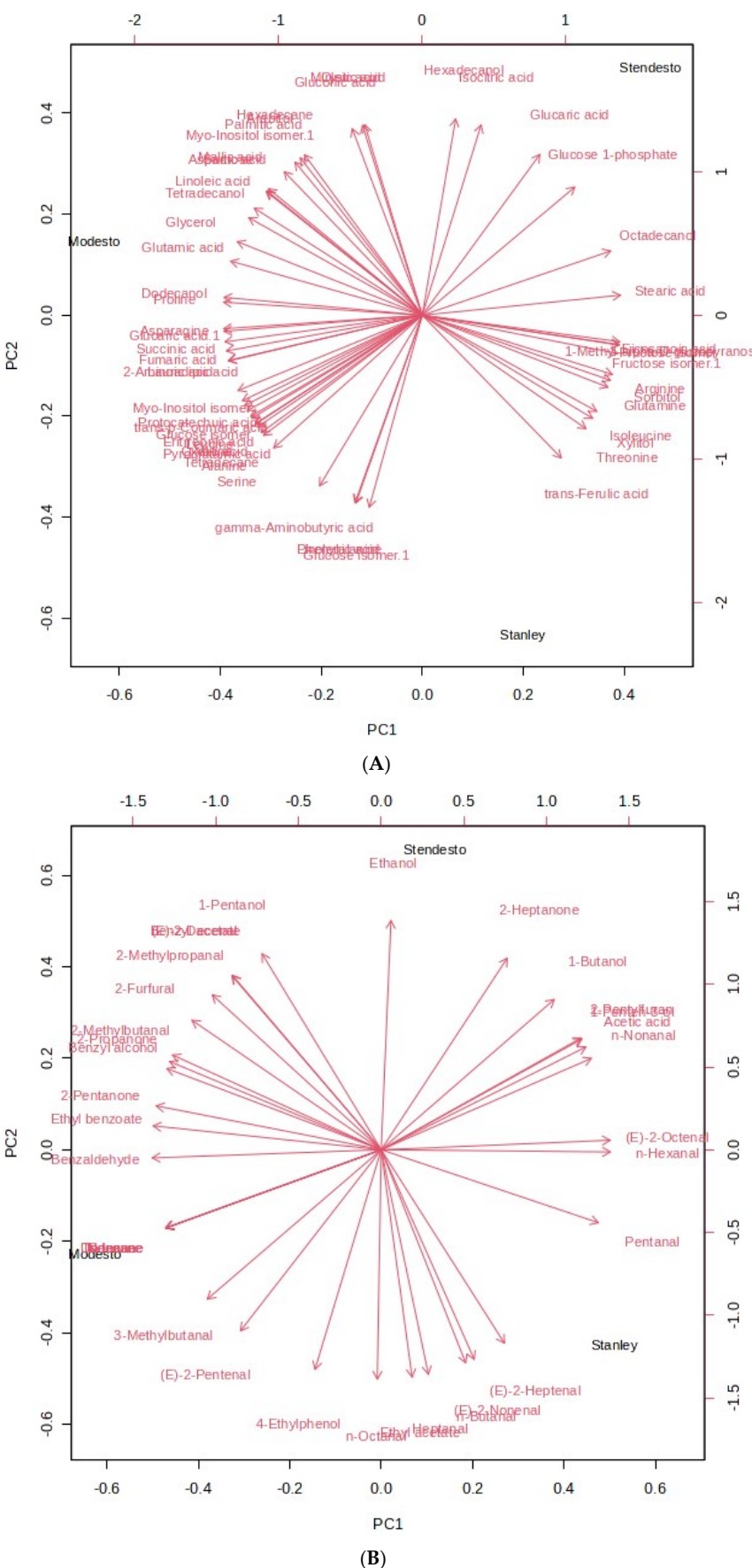

**Figure 4.** Principal component analysis of kernel samples. Eigenvector loading values of compounds: (**A**) primary metabolites; (**B**) VOCs.

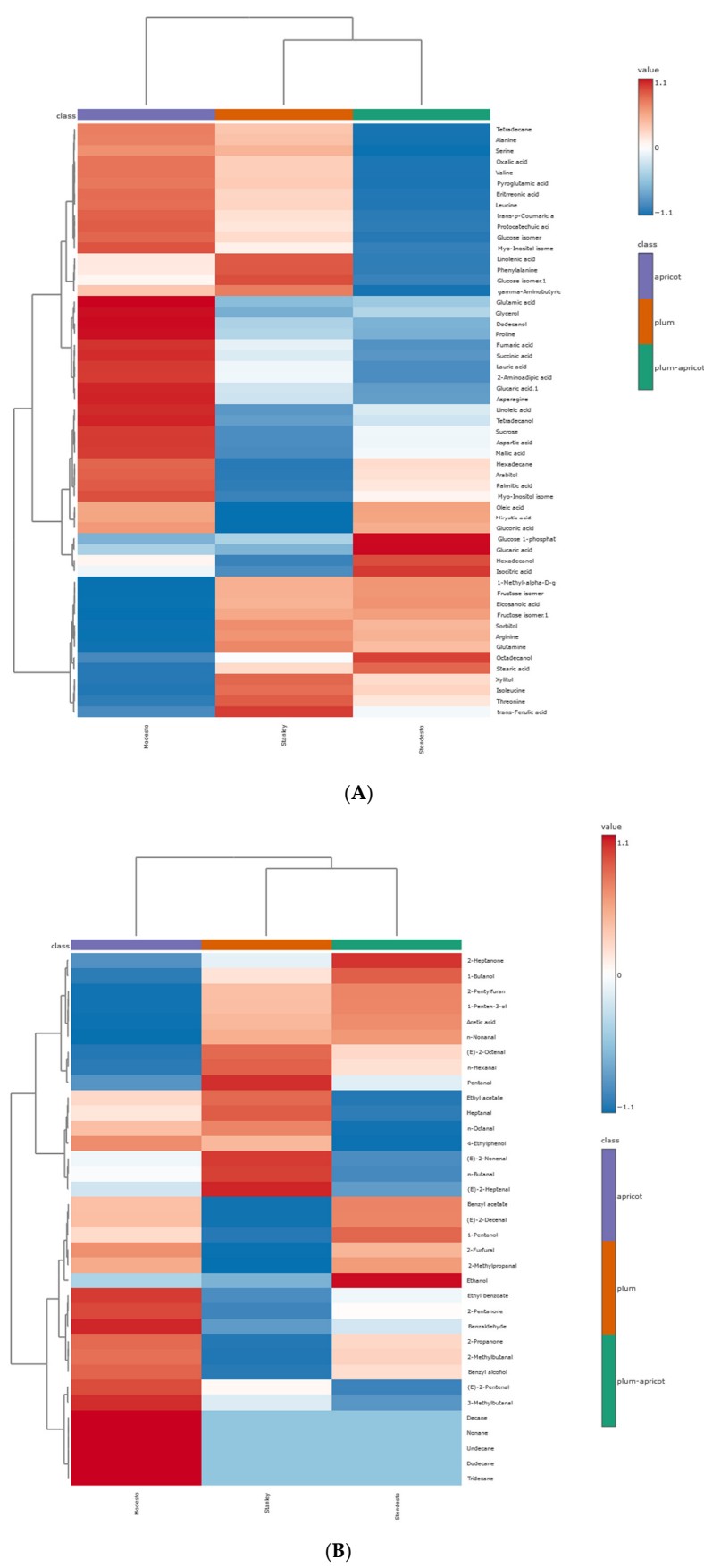

(**A**)

(**B**)

**Figure 5.** Clustering results of kernel samples, shown as heatmap: (**A**) primary metabolites; (**B**) VOCs. The values were normalized by log$_{10}$ transformation.

The apricot kernel is placed separately from the plum and plum–apricot hybrid kernels. The results from the PCA and HCA were useful to preliminarily distinguish the samples. A correlation analysis of the data is presented in Figure 6. It is assumed that the positively correlated metabolite pairs have similar chemical composition, biological function, and homogeneous characteristics [47].

A positive correlation has been established between n-hexanal and pentanal, n-nonanal, acetic acid, 1-butanol, 2-pentylfuran, 1-penten-3-ol, and (E)-2-octenal. Benzaldehyde was positively correlated with sixteen structures, including ethyl benzoate, tridecane, undecane, decane, and benzyl alcohol, among others. Additionally, benzyl alcohol had a positive correlation with fifteen VOCs (2-propanone, 2-furfural, ethyl benzoate, benzyl acetate, and tridecane, among others). Serine and nineteen of the identified metabolites (alanine, valine, leucine, oxalic acid, and fumaric acid, among others) had a positive correlation. Mallic acid was positively correlated with sucrose, aspartic acid, linoleic acid, palmitic acid, oleic acid, and eight others. Sorbitol has a positive correlation with eleven metabolites, including arginine, glutamine, isoleucine, eicosanoic acid, xylitol, threonine, and others. Oleic acid was positively correlated with fifteen metabolites (myristic and gluconic acids having the highest correlation values).

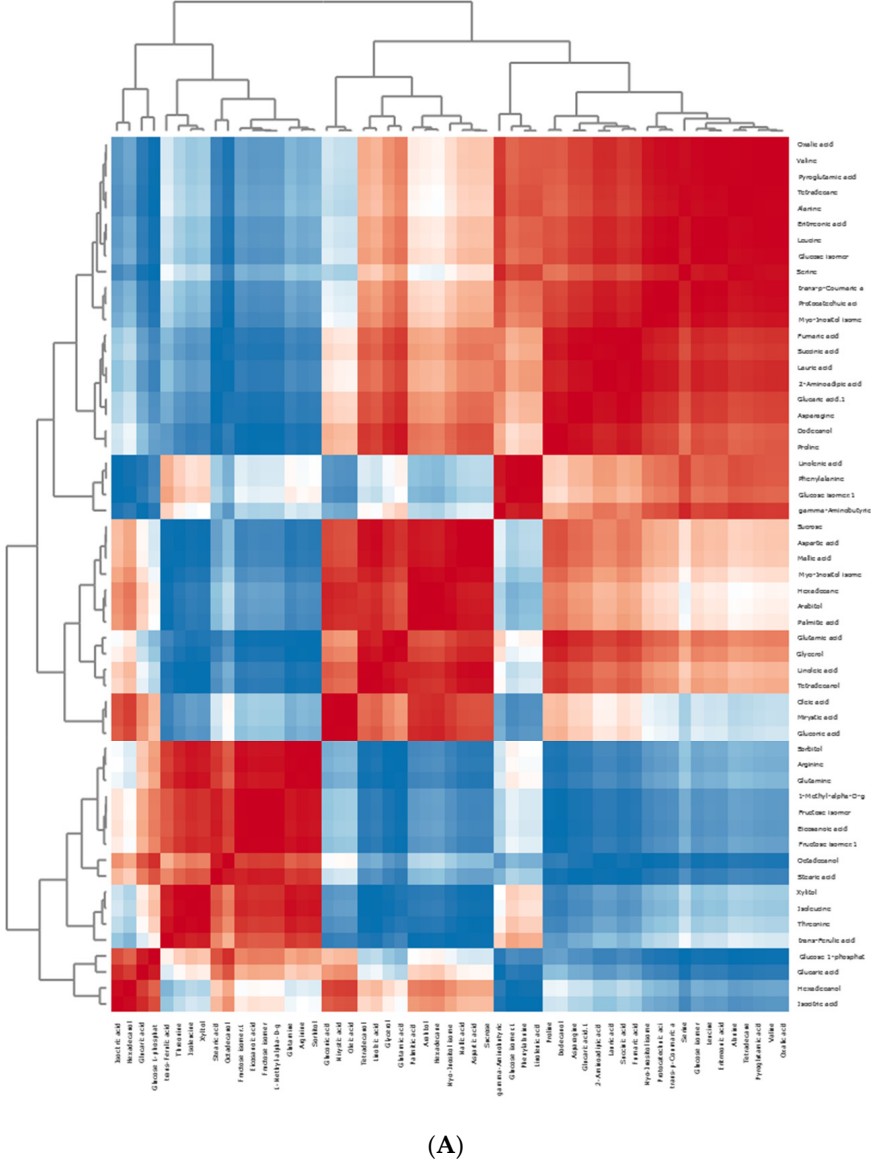

**(A)**

**Figure 6.** *Cont.*

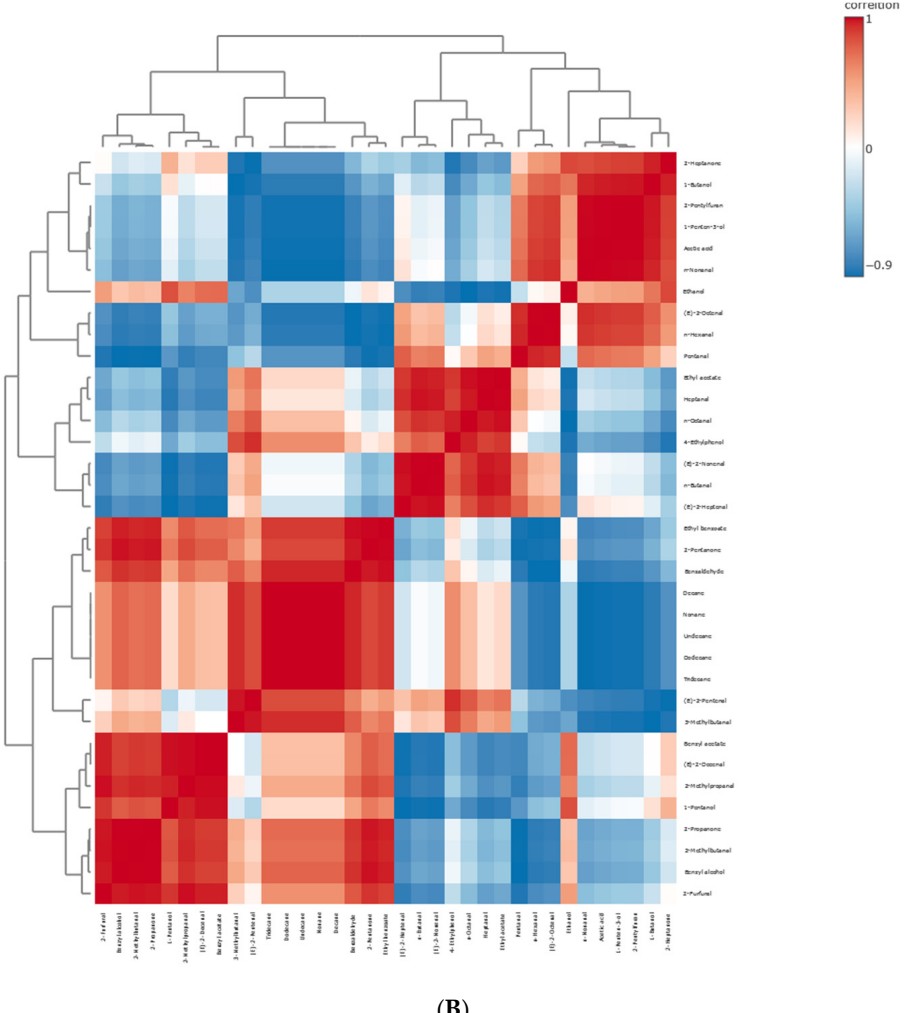

(**B**)

**Figure 6.** Pearson's correlation heatmaps of the differential compounds in studied kernels: (**A**) primary metabolites; (**B**) VOCs. The values were normalized by $\log_{10}$ transformation.

## 3. Materials and Methods

### 3.1. Materials

Three kernels from the following fruit varieties were objects of analysis: "Modesto" (apricot), "Stanley" (plum), and "Stendesto" (plum–apricot hybrid). The kernels were collected from several trees of the same variety. All trees were part of the same plantation. A total of sixty kernels per variety ("Modesto", "Stendesto", and "Stanley") were gathered and thoroughly cleaned from the pulp. All kernels were extracted from ripe fruits. The kernels were left to air-dry, and then were broken into pieces with a Bosch MCM3PM386 robot (Robert Bosch GmbH, Germany) and after that ground with the aid of WMF 0417070011 grinder (Tefal OBH Nordica Group AB, Sweeden). Each kernel variety was placed in a sterile container and kept in a cool, dark, and dry place prior to analysis.

### 3.2. Headspace Solid-Phase Microextraction (HS-SPME) and Gas Chromatography–Mass Spectrometry Analysis (GC–MS)

A 2 cm SPME fiber assembly Divinylbenzene/Carboxen/Polydimethylsiloxane (DVB/CAR/PDMS, Supelco, Bellefonte, PA, USA) was utilized for headspace sampling.

The HS-SPME extraction technique of the studied kernels followed the description of Uekane et al. [48]. An online integrated sampling procedure was automatically performed with a G1888 Network Headspace Sampler. An Agilent 7890A GC unit coupled to an Agilent 5975C MSD and a DB-5ms (30 m × 0.25 mm × 0.25 μm) column were used to

analyze the volatile compounds in all samples. The oven temperature program was the following: from 40 °C (hold 1 min) to 250 °C (hold 5 min) at 2 °C/min; carrier gas: helium with flow rate: 1.0 mL/min; transfer line temperature: 270 °C; ion source temperature: 200 °C, EI energy: 70 eV, mass range: 50 to 550 m/z at 1.0 s/decade.

The extraction procedure of the polar and lipid fractions was completed as described by Ivanova et al. [49]: 0.05 g freeze-dried material was mixed with 1.0 mL methanol/water (75:25, *v/v*) solution and 50.0 μL of each internal standard (nonadecanoic acid methyl ester, ribitol, each in concentration 1.0 mg/mL for the quantification of metabolites of fractions A, and B, respectively), followed by heating at 70 °C for 1 h in a laboratory thermomixer (Analytik Jena AG). The solution, cooling to room temperature, was subjected to the following procedure: 500.0 mL chloroform and 200.0 mL water were added, and then the mixture was centrifuged (5 min/22 °C/13,000 rpm). The lower phase was designed for the analysis of non-polar substances (fraction A), whereas the upper phase for the polar constituents (fraction B). The two phases obtained were vacuum-dried in a centrifugal vacuum concentrator (Labconco Centrivap) at 40 °C. To the dried residue of fraction "A", 1.0 mL 2% $H_2SO_4$ in methanol were added and the mixture was heated on Thermo-Shaker TS-100 (1 h/96 °C/300 rpm). After cooling, the solution was extracted with n-hexane (3 × 500.0 mL). Combined organic layers were vacuum-dried in a centrifugal vacuum concentrator (Labconco Centrivap) at 40 °C.

Prior to the gas chromatography–mass spectrometry (GC–MS) analysis, fractions "A" and "B" were derivatized by the following procedures: 100.0 μL pyridine and 100.0 μL BSTFA were added to the dried residue (fraction "A"), then heated on Thermoshaker, Analytik Jena AG, Germany (45 min/70 °C/300 rpm). 1.0 μL from the solution was injected into the GC–MS.; 300.0 μL solution of methoxyamine hydrochloride (20.0 mg/mL in pyridine) was added to dried residue (fraction "B"), and the mixture was heated on Thermo-Shaker TS-100 (1 h/70 °C/300 rpm). After cooling, 100.0 μL BSTFA were added to the mixture then heated on Thermoshaker, Analytik Jena AG, Germany (40 min/70 °C/300 rpm), and 1.0 μL from the solution was injected into the GC–MS system.

The 2.73 AMDIS software (Automated Mass Spectral Deconvolution and Identification System, NIST, Gaithersburg, MD, USA) assisted in the reading of the mass spectra and the metabolite identification. The separated compounds were compared to the GC–MS spectra and retention indices (RI) of reference compounds in the Golm Metabolome Database (http://csbdb.mpimp-golm.mpg.de/csbdb/gmd/gmd.html, accessed on 1 December 2023) and NIST'08 database (NIST Mass Spectral Database, PC-Version 5.0, 2008 from National Institute of Standards and Technology, Gaithersburg, MD, USA). The 2.73 AMDIS software recorded the RIs of the compounds with a standard n-hydrocarbon calibration mixture ($C_{8-36}$, Restek, Teknokroma, Spain). Analyses were triplicated for each kernel variety.

### 3.3. Statistical Analysis

MS Excel software 365 was used for data analysis. Results are presented as mean ± SD (standard deviation). Additional statistical analyses of the data were presented using one-way ANOVA and a Tukey–Kramer post hoc test ($\alpha$ = 0.05), as described by Assaad et al. [50]. PCA and HCA of GC–MS data were conducted using MetaboAnalyst, a web-based platform (www.metaboanalyst.ca, accessed on 15 December 2023). The concentrations of the identified compounds were employed for PCA. All zero values were replaced with a value (1/2 of the minimum positive values in the original data) assumed to be the detection limit. PCA (95% confidence level) was employed to calculate the eigenvector loading values and to identify the major statistically different components among the samples. The GC–MS data were also subjected to HCA, which produced a Ward dendrogram of hierarchical clustering and Euclidean distance measurement between the analyzed samples. The values were normalized by $\log_{10}$ transformation.

## 4. Conclusions

No data regarding the primary metabolites and VOCs of kernels from the "Modesto" (apricot), "Stanley" (plum), and "Stendesto" (plum–apricot) varieties were present in literature, which makes this a first comprehensive evaluation regarding this important byproduct. In total, fifty-five metabolites were identified belonging to the following chemical groups: phenolic acids, hydrocarbons, organic acids, fatty acids, sugar acids and alcohols, mono- and disaccharides, and amino acids. The most abundant were the fatty acids, sugar acids and alcohols, and mono- and disaccharides. The hybrid kernel generally inherited all the metabolites present in the parental kernels. Thirty-five VOCs were identified from the three samples, with aldehydes contributing most. Considering the VOCs, the hybrid kernel had more resemblance to the plum one, bearing that alkanes were only identified in the apricot kernel.

The applied PCA placed the plum and plum–apricot kernels in the same group, leaving the apricot kernel in a separate group. The obtained results can successfully be used as a reference and stepping stone for future analyses. Focusing attention on kernels as potential nutritional and functional sources will definitely aid with the utilization of this by-product.

**Author Contributions:** Conceptualization, D.M. and A.P.; methodology, I.D.; software, I.D.; validation, D.M., A.P. and I.D.; formal analysis, I.D.; investigation, D.M., S.P. and A.P.; resources, S.P.; data curation, I.D.; writing—original draft preparation, A.P., D.M., I.D. and S.P.; writing—review and editing, D.M., A.P., I.D. and S.P.; visualization, A.P.; supervision, D.M.; project administration, A.P.; funding acquisition, A.P. All authors have read and agreed to the published version of the manuscript.

**Funding:** This research and the APC were funded by the Bulgarian National Science Fund, project no. КП-06-Н67/2.

**Institutional Review Board Statement:** Not applicable.

**Informed Consent Statement:** Not applicable.

**Data Availability Statement:** The data presented in this study are available on request from the corresponding author. The data are not publicly available due to privacy

**Acknowledgments:** The authors would like to acknowledge Argir Zhivondov for actively working on expanding plumcot varieties in Bulgaria, and registering the "Stendesto".

**Conflicts of Interest:** The authors declare no conflicts of interest. The funders had no role in the design of the study; in the collection, analyses, or interpretation of data; in the writing of the manuscript; or in the decision to publish the results.

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
