# Peer review of "HS-SPME-GC–MS Profiling of Volatile Organic Compounds and Polar and Lipid Metabolites of the “Stendesto” Plum–Apricot Kernel with Reference to Its Parents"

_horticulturae, doi:10.3390/horticulturae10030257_

Round 1

Reviewer 1 Report

Comments and Suggestions for Authors

Here are my suggestions for authors in order to improve manuscript quality.

All are listed below with an appropriate Line number(s) from text in order to facilitate tracking:

Lines 6-8: Please correct typos- words "Food" and "Technology" in your affiliations should be given with capital letters.

Lines 15-17: Please specify the aim of your research. Abstract should be self-explanatory.

Line 16: "for reason" instead of "by reason". Correct.

Line 20: "compounds" instead of "structures" here.

Line 23: typo - I think it should be "thirty-five" here? Check/correct.

Line 43: "metabolites" in plural here. Also, define "VOC" abbreviation. This is the first mention in the main text.

Line 49: I think that "experimental chemistry" is better term here instead of "scientific chemistry".

Line 60: Please specify here these "Scientific papers" mentioned, provide references.

Line 72: typo - references should be listed before the end of sentence here, not after.

Lines 83-84 and 95-96: Similar typos like above one. Move cited references to be included in the previous sentences.

Line 110: Check sentence. I think that something is missing here after "and" in front of "were"?

Lines 112-113: Please correct all decimal numbers in the Table to be presented with decimal point instead of decimal comma, to be in the line with English grammar. This error is repeating in all other Tables. Please revise all. Also, in the Table 1 you have to times "Glucaric acid" presented. Check/correct. Also, "alpha" should be given as Greek symbol, as well as word "trans" and label "p-" should be given in Italic to be in line with modern Organic chemistry nomenclature. Please correct all these issues.

Line 127: Please rewrite this Line. It is incomprehensible to me.

Line 140: Same comment for "p-trans" as in the Table 1.

Lines 152-153: Suggest to rewrite as follow: "Due to the lack of relevant research comparison with subject reference to..."

Line 154: "compounds" instead of "structures".

Lines 157-159: I would like to suggest to authors to calculate UFA/SFA ratio as useful nutritional parameter to see how fit your kernels in suggested range. According to literature UFA/SFA ratio should be higher than 1.6 to be favorable.

Line 184: Please define TIC abbreviation. This is the first mention in the text.

Line 207: typo - no capital letter in "2-heptanone" name here.

Line 226: I think it should be here "In terms of..."? Check/correct.

Line 229: I would reorder the listed cultivars to present firstly parents and than hybrid.

Kind regards.

Comments on the Quality of English Language

Some minor corrections, suggested in comments for authors, are needed.

Author Response

The authors would like to show their appreciation for the reviewer’s helpful review. We accept his\her suggestions and improvements for the manuscript.

Below, the reviewer can find a point by point answer to all comments:

Lines 6-8: Please correct typos- words "Food" and "Technology" in your affiliations should be given with capital letters. – a correction has been made.

Lines 15-17: Please specify the aim of your research. Abstract should be self-explanatory. – a revision of the abstract has been made.

Line 16: "for reason" instead of "by reason". Correct. – a change has been made

Line 20: "compounds" instead of "structures" here. – a change has been made

Line 23: typo - I think it should be "thirty-five" here? Check/correct. – checked and corrected to thirty-five

Line 43: "metabolites" in plural here. Also, define "VOC" abbreviation. This is the first mention in the main text. The abbreviation has been introduced, and a change in plural form has been made, although we think it should be singular since it applies to composition.

Line 49: I think that "experimental chemistry" is better term here instead of "scientific chemistry". Corrected as requested

Line 60: Please specify here these "Scientific papers" mentioned, provide references. – reference added

Line 72: typo - references should be listed before the end of sentence here, not after. The reference in L72 is listed where the sentence ends and we do not see a problem. Probably we cannot fully understand the suggestion here. We noticed a mishap with the software used to produce the referencing. We have updated the list and we believe that referencing is correct and where it is supposed to appear.

Lines 83-84 and 95-96: Similar typos like above one. Move cited references to be included in the previous sentences. – changes made, references appear at the end of relevant sentences.

Line 110: Check sentence. I think that something is missing here after "and" in front of "were"? – checked and corrected

Lines 112-113: Please correct all decimal numbers in the Table to be presented with decimal point instead of decimal comma, to be in the line with English grammar. This error is repeating in all other Tables. Please revise all. Also, in the Table 1 you have to times "Glucaric acid" presented. Check/correct. Also, "alpha" should be given as Greek symbol, as well as word "trans" and label "p-" should be given in Italic to be in line with modern Organic chemistry nomenclature. Please correct all these issues. – checked and corrected

Line 127: Please rewrite this Line. It is incomprehensible to me. – correction made

Line 140: Same comment for "p-trans" as in the Table 1. - corrected

Lines 152-153: Suggest to rewrite as follow: "Due to the lack of relevant research comparison with subject reference to..." done

Line 154: "compounds" instead of "structures". done

Lines 157-159: I would like to suggest to authors to calculate UFA/SFA ratio as useful nutritional parameter to see how fit your kernels in suggested range. According to literature UFA/SFA ratio should be higher than 1.6 to be favorable. – the ratio has been calculated, information is present in L159-160

Line 184: Please define TIC abbreviation. This is the first mention in the text. - done

Line 207: typo - no capital letter in "2-heptanone" name here. – this is the beginning of the sentence, the capital is needed since the number cannot be capitalized.

Line 226: I think it should be here "In terms of..."? Check/correct. corrected

Line 229: I would reorder the listed cultivars to present firstly parents and than hybrid. Re-ordered as requested

We very much hope that that with the answers and clarification being provided the improved manuscript will be accepted for publication.

Reviewer 2 Report

Comments and Suggestions for Authors

Comments:
The study titled “HS-SPME-GC-MS profiling of kernel primary metabolites to reveal the relationship between a plum-apricot hybrid and its parents" by Mihaylova et al. is interesting. It assesses the metabolic profile of the kernels derived from a plum-apricot hybrid, Stendesto, and its parents Modesto (apricot) and Stanley (plum). The data presented in the study could be useful for the plum/apricot breeders and consumers. However, it posits points that need to be addressed. Authors are kindly suggested to do so for the comments provided below.

Major comments:

1. The study title implies that the study carries out the metabolites-based relationship assessment of the hybrid to its parents. However, it largely discusses on the metabolite data without discussing their relevance to their relationship. Authors are suggested to either modify the title or modify the main-text to satisfy the title.

2. Neither of the bar charts presented in the study constitutes any statistical significance assessment. Authors are requested to use suitable statistical significance tests for them.

3. The study would benefit greatly if the authors were to emphasize on the significance of the data presented. Besides being one of the few new studies on the hybrids, the study may offer additional significance on research and breeding. Authors are suggested to make amends accordingly.

4. While the authors have often emphasized on lack of relevant prior studies to make comparison, there are some studies- even though they may specifically be on the species/varieties used in the study. Comparing and contrasting study findings with those studies is expected to greatly benefit the study. Authors are suggested to do so.

5. Were the kernels harvested from the ripened fruits? Please clarify the case in the materials and method section.

Minor comments:
1. The manuscript constitutes some minor typos and grammatical errors (e.g., Page 10; lines 257-258: n-nonanal => n-nonanol (?)). Authors are advised to have it proofread by someone professionally proficient in English.

Comments on the Quality of English Language

Author Response

The authors would like to show their appreciation for the reviewer’s helpful review. We accept his\her suggestions and improvements for the manuscript.

Below, the reviewer can find a point by point answer to all comments:

Comment: 1. The study title implies that the study carries out the metabolites-based relationship assessment of the hybrid to its parents. However, it largely discusses on the metabolite data without discussing their relevance to their relationship. Authors are suggested to either modify the title or modify the main-text to satisfy the title. Answer: The authors are suggesting a modified version of their initial title.

Comment: 2. Neither of the bar charts presented in the study constitutes any statistical significance assessment. Authors are requested to use suitable statistical significance tests for them. Answer: Statistical significance has been added to relevant figures.

Comment: 3. The study would benefit greatly if the authors were to emphasize on the significance of the data presented. Besides being one of the few new studies on the hybrids, the study may offer additional significance on research and breeding. Authors are suggested to make amends accordingly. Answer: We have included this suggestion in the conclusion section according to comment.

Comment: 4. While the authors have often emphasized on lack of relevant prior studies to make comparison, there are some studies- even though they may specifically be on the species/varieties used in the study. Comparing and contrasting study findings with those studies is expected to greatly benefit the study. Authors are suggested to do so. Answer: We agree with the reviewer. As we have mentioned in the introduction a major setback in research is the lack to variety/cultivar citation in papers. We believe that this makes comparison rather difficult. We have tried to include some comparison about apricot kernels, since few papers on the subject can be found.

Comment: 5. Were the kernels harvested from the ripened fruits? Please clarify the case in the materials and method section. Answer: Kernels were harvested from ripe fruits. This is now added to the MM section as required.

Comment: 1. The manuscript constitutes some minor typos and grammatical errors (e.g., Page 10; lines 257-258: n-nonanal => n-nonanol (?)). Authors are advised to have it proofread by someone professionally proficient in English. Answer: The manuscript has been checked and appropriate corrections have been made.

We very much hope that that with the answers and clarification being provided the improved manuscript will be accepted for publication.

Reviewer 3 Report

Comments and Suggestions for Authors

The authors investigated the kerneI primary metabolites of plum-apricot hybrid and its parents using HS-SPME-GC-MS to reveal their relationship. I have thoroughly read the manuscript and I don’t think the manuscript should be published in this journal in the current version.

Primary metabolites are substances for maintaining the growth and reproduction, such as amino acids, nucleotides, polysaccharides, lipids, vitamins, etc. The authors told the readers they investigated primary metabolites of plum-apricot hybrid and its parents, however, they actually examined amino acids, organic acids, sugar acid and alchols, saccharides, phenolic acids, fatty acids, and volatile compounds. I suggest that the authors improve the title of the manuscript.

Currently, UPLC-MS coupled with commercial or non-commercial databases is widely used for the determination and identification of amino acids, organic acids, and other non-volatile compounds. Why do the authors use GC-MS?

Please tell the readers the relationship between your study and trending topics like zero-waste management.

Why do the authos conduct this study? What is the knowledge gap between your study and the literatures? Only “No data about primary metabolites and VOCs of kernels from the “Modesto” (apri-340 cot), “Stanley” (plum), and “Stendesto” (plum-apricot) varieties was present in literature” mentioned in the conclusions section?

Please check the symbol of Celsius degree.

Please provide the full name of the abbreviation when it appears in the manuscript for the first time.

Please improve the qulity of figure 3, figure 4, and figure 5. I can’t see anythig in these figures.

Comments on the Quality of English Language

Minor editing of English language required

Author Response

The authors would like to show their appreciation for the reviewer’s helpful review. We accept his\her suggestions and improvements for the manuscript.

Below, the reviewer can find a point by point answer to all comments:

Comment: Primary metabolites are substances for maintaining the growth and reproduction, such as amino acids, nucleotides, polysaccharides, lipids, vitamins, etc. The authors told the readers they investigated primary metabolites of plum-apricot hybrid and its parents, however, they actually examined amino acids, organic acids, sugar acid and alchols, saccharides, phenolic acids, fatty acids, and volatile compounds. I suggest that the authors improve the title of the manuscript. Answer: The authors have provided an updated version of the title.

Comment: Currently, UPLC-MS coupled with commercial or non-commercial databases is widely used for the determination and identification of amino acids, organic acids, and other non-volatile compounds. Why do the authors use GC-MS? Answer: Generally, gas chromatography/mass spectrometry (GC/MS) is used to analyse the primary metabolites (organic acids, amino acids, sugars and sugars alcohols) and nonpolar compounds (fatty acids and sterols). The main advantages of using GC-MS for metabolomics are its high chromatographic separation power, high peak capacity, reproducible retention times, robust quantitation, high selectivity and sensitivity, and fast compound identification using existing commercial (e.g., NIST) and open source (e.g., The Golm Metabolome Database) spectral libraries. Also, unlike LC-MS, GC-MS generates reproducible molecular fragmentation patterns, making it an integral tool for metabolite identification. With GC-MS, the sample is often ionized by electron ionization (EI), which are stable, reproducible and compound-specific.

Comment: Please tell the readers the relationship between your study and trending topics like zero-waste management. Answer: We believe that highlighting obvious by-products like kernels as potential nutritional and functional sources is a trending topic directly linked to zero-waste management. Some additional information is not added to clear the authors initial thoughts.

Comment: Why do the authos conduct this study? What is the knowledge gap between your study and the literatures? Only “No data about primary metabolites and VOCs of kernels from the “Modesto” (apri-340 cot), “Stanley” (plum), and “Stendesto” (plum-apricot) varieties was present in literature” mentioned in the conclusions section? Answer: The authors have stated in their research aim that as far as their knowledge is concerned this is a pioneer study on the topic of kernel characterization. We believe that we have stated this information not only in the Conclusion section.

Comment: Please check the symbol of Celsius degree. Answer: Checked

Comment: Please provide the full name of the abbreviation when it appears in the manuscript for the first time. Answer: abbreviations have been introduced in the manuscript where they first appear, thank you for this remark.

Comment: Please improve the qulity of figure 3, figure 4, and figure 5. I can’t see anythig in these figures. Answer: The authors have used the Metaboanalyst software which provides the figures for download in picture formats (jpeg, tiff, among some others). We cannot make any changes to them not to the font nor the size of the data. The sole thing we can do is to enlarge them so that they can become visibly bigger. Currently, we are not provided with any better opportunity to present such results.

We very much hope that that with the answers and clarification being provided the improved manuscript will be accepted for publication.

Reviewer 4 Report

Comments and Suggestions for Authors

In the manuscript that was submitted by Dasha Mihaylova, Aneta Popova and colleagues, entitled "HS-SPME-GC-MS profiling of kernel primary metabolites to reveal the relationship between a plum-apricot hybrid and its parents" the authors study the kernal metabolic profile of plum-apricot hybrid against plum and apricot. The results indicate that the hybrid metabolic profile is closer to the plum parent. Overall, the writing is moderate with several grammar mistakes, and the figures need improvement. The current manuscript is in line with the aims and purposes of the Horticulturae journal, however there are several issues that should be carefully addressed.

Major and minor

Abstract

Lines 15-17. The authors mention ‘Plums and apricots as well-known and preferred by consumers by reason of their distinct sensory and health beneficial properties.’ Of course, you mean the fruits, but what about the kernels, which are consumed by the public?

Lines 27-29. The authors should describe the results regarding PCA and hierarchical analysis and not just mention them.

Lines 26-27. This should be the last sentence.

Introduction

Line 35. Replace matrices with plant tissues and organs.

What about amygdalin and other compounds that are found in plum and apricot kernel. A more extensive introduction about kernels and more obvious almond kernel should be done.

Results and discussion section

An obvious metabolic comparison among apricots, plums, and hybrids could be in the flesh of fruit. Why did you choose to present us with the kernels alone? What about the flesh of these cultivars?

How exactly do you perform odor description. Describe in MM section.

Fig. 1. Delete the 2 decimals from y axis. Error bars are missing. Start with Uppercase letter. Amino acids and so on. The description of the legend should be clear enough to understand. For instance, post-hoc test among cultivars is missing.

Fig. 2. Like Fig. 1

Fig. 3. At least display the percentage in each pie.

Fig. 4. Incorporate percentage in PC1 and PC2, and increase the size of the font.

Fig. 5 and 6. increase the size of the font.

Comments on the Quality of English Language

line 16 as replace with are and by reason of replace with because of

line 20 structures, namely, and so on 

line 22 studied replace with kernel

line 23 profiled replace with identified 

This sentence does not make sense 'This study is the first providing information about the metabolite profile of not only kernels, but also hybrids.'

All these are only in the abstract. The rest of the manuscript is better but certainly need improvement.

Author Response

The authors would like to show their appreciation for the reviewer’s helpful review. We accept his\her suggestions and improvements for the manuscript.

Below, the reviewer can find a point by point answer to all comments:

Comment: Lines 15-17. The authors mention ‘Plums and apricots as well-known and preferred by consumers by reason of their distinct sensory and health beneficial properties.’ Of course, you mean the fruits, but what about the kernels, which are consumed by the public? Answer: Some information about the consumption of kernels has been added to the abstract.

Comment: Lines 27-29. The authors should describe the results regarding PCA and hierarchical analysis and not just mention them. Answer: The authors have provided an updated version of the abstract. We believe that an abstract should be self-explanatory and we agree with the reviewer, but the authors should also comply with the journals recommendations about the number of words used in the abstract, thus we think that not every single result can be included in the abstract.

Comment: Lines 26-27. This should be the last sentence. Answer: The authors have provided an updated version of the abstract.

Comment: Line 35. Replace matrices with plant tissues and organs. Answer: the required correction has been made.

Comment: What about amygdalin and other compounds that are found in plum and apricot kernel. A more extensive introduction about kernels and more obvious almond kernel should be done. Answer: Authors have updated the content on the introduction as requested.

Comment: An obvious metabolic comparison among apricots, plums, and hybrids could be in the flesh of fruit. Why did you choose to present us with the kernels alone? What about the flesh of these cultivars? Answer: The authors intend to present data about fruit flesh in future research. The design of our current manuscript is focused on kernels alone as a possible utilization of otherwise waste.

Comment: How exactly do you perform odor description. Describe in MM section. Answer: We have used a descriptor of volatiles to be able to present them in sections. We have used https://foodb.ca/ as a source (this is now clarified in the manuscript). However, we find it unsuitable to add in the MM section.

Comment: Fig. 1. Delete the 2 decimals from y axis. Error bars are missing. Start with Uppercase letter. Amino acids and so on. The description of the legend should be clear enough to understand. For instance, post-hoc test among cultivars is missing. Answer: The figure has been updated as requested.

Comment: Fig. 2. Like Fig. 1 Answer: Update has been done accordingly.

Comment: Fig. 3. At least display the percentage in each pie. Answer: We strongly believe that this is a presentation decision. We have added relevant information in the text. We believe that adding % will not make the figure more comprehensible.

Comment: Fig. 4. Incorporate percentage in PC1 and PC2, and increase the size of the font. Answer: Unfortunately, we cannot make the required changes. As we have stated we have used the Metaboanalyst software, where we can download the data as a ready non-editable picture. Currently, we are not provided with better software to present such results.

 Comment: Fig. 5 and 6. increase the size of the font. Answer: Unfortunately, we cannot make the required changes. As we have stated we have used the Metaboanalyst software, where we can download the data as a ready non-editable picture. Currently, we are not provided with better software to present such results.

Comment: line 16 as replace with are and by reason of replace with because of Answer: a change has been made

Comment: line 20 structures, namely, and so on Answer: the authors have made a change in this sentence, but we have not combined the two sentences in one. We believe that longer sentences are not always better. Although it is possible to combine the sentences in question.

Comment: line 22 studied replace with kernel Answer: the requested change has been made.

Comment: line 23 profiled replace with identified Answer: we strongly believe that this is a personal choice of words, but the change has been made. Additionally, by making this change, we would like to point out that a repetition of words occurs.

Comment: This sentence does not make sense 'This study is the first providing information about the metabolite profile of not only kernels, but also hybrids.' Answer: We think the sentence is grammatically correct, but since the reviewer find it hard to understand we have made some changes.

We very much hope that that with the answers and clarification being provided the improved manuscript will be accepted for publication.

Round 2

Reviewer 2 Report

Comments and Suggestions for Authors

Comments:

The authors have made some changes to the manuscript. There are still some parts that can be improved, which are listed below. Authors are kindly suggested to address them.

1.  Since the authors apparently carried out the element (compound/metabolite) comparison among three species during statistical analysis, the bar-graph would be more reader-friendly if the bars were to be clustered as per element (compound/metabolite) and color-code the species instead of clustering the color-coded elements in each species.

2.  Figure 2 bar chart: Statistical comparison marker is missing for Modesto Alkanes

Author Response

The authors would like to show their appreciation for the reviewer and his/her continuous effort in the review process.

We have made the suggested changes concerning the figure.

Concerning Figure 2 the element is not missing, Alkanes were only identified in the Modesto variety, thus we cannot have any statistical evaluation there apart from ±SD.

We very much hope that the revised manuscript will be seen as worthy of publishing.

Reviewer 3 Report

Comments and Suggestions for Authors

Accepted as it is

Author Response

The authors would like to show their appreciation for the reviewer and his/her continuous effort in the review process. Thank you for accepting the manuscript for publication.

Reviewer 4 Report

Comments and Suggestions for Authors

Based on the answer of the authors, they should incorporate this answer that focuses in waste like kernels in the text. 'Comment: An obvious metabolic comparison among apricots, plums, and hybrids could be in the flesh of fruit. Why did you choose to present us with the kernels alone? What about the flesh of these cultivars? Answer: The authors intend to present data about fruit flesh in future research. The design of our current manuscript is focused on kernels alone as a possible utilization of otherwise waste.' Except for that, all the other concerns have been successfully addressed; hence, I recommend this manuscript for publication in the Horticulturae Journal. 

Author Response

The authors would like to thank the reviewer for finding the time to re-evaluate the manuscript. We have included our previous answer in the manuscript.

We would also like to thank the reviewer for recommending the manuscript for publication.